# Language-based Action Concept Spaces Improve Video Self-Supervised Learning

**Kanchana Ranasinghe**
Stony Brook University
kranasinghe@cs.stonybrook.edu

**Michael Ryoo**
Stony Brook University
mryoo@cs.stonybrook.edu

## Abstract

Recent contrastive language image pre-training has led to learning highly transferable and robust image representations. However, adapting these models to video domain with minimal supervision remains an open problem. We explore a simple step in that direction, using language tied self-supervised learning to adapt an image CLIP model to the video domain. A backbone modified for temporal modeling is trained under self-distillation settings with train objectives operating in an *action concept space*. Feature vectors of various action concepts extracted from a language encoder using relevant textual prompts construct this space. A large language model aware of actions and their attributes generates the relevant textual prompts. We introduce two train objectives, *concept distillation* and *concept alignment*, that retain generality of original representations while enforcing relations between actions and their attributes. Our approach improves zero-shot and linear probing performance on three action recognition benchmarks.

## 1   Introduction

Actions in videos are defined by individual objects, their relationships, and interaction [1, 2]. Video self-supervised learning focuses on discovering representations aware of such action attributes directly from video content with no human supervision [3]. Particularly in the case of videos, where manual human annotation can be both expensive and noisy, such self-supervised approaches are invaluable.

A recent variant of self-supervision explores learning with loosely paired image-caption pairs, leading to highly transferable and robust representations such as CLIP [4]. These approaches obtain zero-shot performance often comparable to fully-supervised methods. However, their counterparts in the video domain [5, 6, 7, 8, 9, 10, 11] do not exhibit the same generality. In fact, some approaches training CLIP on videos [11, 12] perform subpar to image-CLIP under zero-shot settings (see Table 2). Such behaviour can be attributed to lesser availability and more noisy nature of labelled (or paired caption) video datasets [3]. This motivates exploration into self-supervised learning (SSL) techniques that can learn from videos under less supervision while utilizing existing image CLIP [4] like representations. Existing state-of-the-art video SSL approaches [13, 14] learn highly transferable representations from videos, but combining these with image CLIP representations is not straightforward. In fact, despite methods like SVT [13] being able to utilize image SSL representations [15] for weight initialization to achieve better performance, using image CLIP representations instead for weight initialization leads to performance subpar to image CLIP (see Table 4). This raises necessity for alternate video SSL approaches compatible with CLIP like image representations and is our key motivation.

In this work, we explore self-supervised learning techniques that adapt image CLIP models [4] to video domain under entirely self-supervised settings, dependent on no form of video level labels or captions. Under this setting, natural language can still provide strong cues regarding attributes that compose an action category [16, 17]. We leverage this idea to propose a novel *language-based* self-supervised learning objective. Following a standard self-distillation and multi-view based SSL

formulation [15, 13], we introduce language aligned feature spaces, *action concept spaces*, where our SSL objectives operate. Large-language models (LLMs) [18], given their extensive world knowledge [19, 20], serve as an ideal tool to generate necessary textual concepts for these spaces. We also introduce regularization suitable for our language aligned SSL objective to prevent collapse during training. Our resulting framework is termed *Language-based Self-Supervision*, or LSS.

In contrast to existing video self-supervised learning approaches [13, 14], our proposed LSS retains and improves transferability of image CLIP representations much better (see Tables 1 and 4). Additionally, our language aligned learning framework allows direct zero-shot operation on downstream tasks. Moreover, unlike video CLIP methods with similar zero-shot capabilities [5, 6, 7, 8, 9, 10, 11] that utilize per-video labels / captions for learning, our proposed LSS requires only videos for training.

We summarize our key contributions as follows:

- Self-supervised learning paradigm capable of retaining and improving strengths of CLIP like image representations for video domain operation
- Video specific self-supervised learning objectives, namely *concept distillation* and *concept alignment*, that enforce relations between action categories and their visual attributes
- Novel language-based video self-supervised learning framework operating zero-shot on downstream action classification tasks without requiring per-video labels / captions for training

Experiments on action recognition datasets showcase state-of-the-art performance for our learned representations under linear-probing, standard zero-shot, and transductive zero-shot settings.

## 2 Related Work

**Self-Supervised Learning in Videos** was initially dominated by pretext tasks specific to the video domain [21, 22, 23, 24, 25, 26, 27, 28, 29, 30, 31, 32]. Recently a shift to contrastive losses led to [33, 34, 35, 36, 37, 38] with some variants focused on video specific view generation [39, 40, 41, 42, 13]. An alternate direction has been masked auto-encoders [14]. To the best of our knowledge, existing video self-supervised learning (SSL) approaches operate purely within the visual domain. By video SSL, we refer to methods that utilize only videos with no paired captions (or labels) for each video during training. In contrast, our proposed LSS learns purely from videos in a self-supervised manner, integrating pre-trained language-image models to learn language aligned representations.

**Zero-shot Action Recognition** began with manual attribute and feature selection [43, 44, 45, 46, 47] with later works utilizing action word embeddings [16, 17]. The idea of connecting action categories with elaborate descriptions of those actions, within language embedding spaces [48, 49] has been a next step and is closely related to our work. This idea is also explored in image domain to boost zero-shot performance [50]. While our work is inspired by such approaches, in contrast, we use relations between such actions and descriptions as self-supervised signals for learning. Recent image CLIP models [4, 51] are another line of works achieving strong performance on some video classification tasks, with only single frame processing. Multiple approaches build on image CLIP [4] to learn video level representations [11, 52, 53, 54, 12, 55] under fully-supervised settings. While achieving strong performance on the training datasets, their zero-shot improvements over CLIP are minimal or even subpar (see Table 2). Therein, LSS focuses on zero-shot performance under self-supervised settings while retaining (and improving) the generality of the representation space.

**Self-training** methods leverage pseudo-labels on unlabeled data [56, 57, 58] for supervised-fashion training. Recently they have been combined with CLIP models for zero-shot operation [59, 60]. While inspired by such self-training approaches, our proposed LSS differs in its continuous feature space self-distillation, language-based relations enforcing, video domain operation, and cross-dataset transfer for zero-shot operation.

**Adapting image-CLIP models to video** under fully-supervised settings has gathered much interest [5, 6, 7, 8, 9, 10]. Expanding backbones for temporal modeling, multi-modal fusion, secondary training objectives, partial parameter updates, and scaling-up data are key ideas explored [55, 10]. In contrast, LSS is a first to operate under self-supervised settings using no video annotations.

**Contemporary work** in [61] adapts image CLIP features to video tasks label free similar to our work. ViFi-CLIP [62] introduces zero-shot action recognition benchmarks and similarly adapts CLIP to videos retaining generality. Using LLMs for action recognition is also explored in [63].

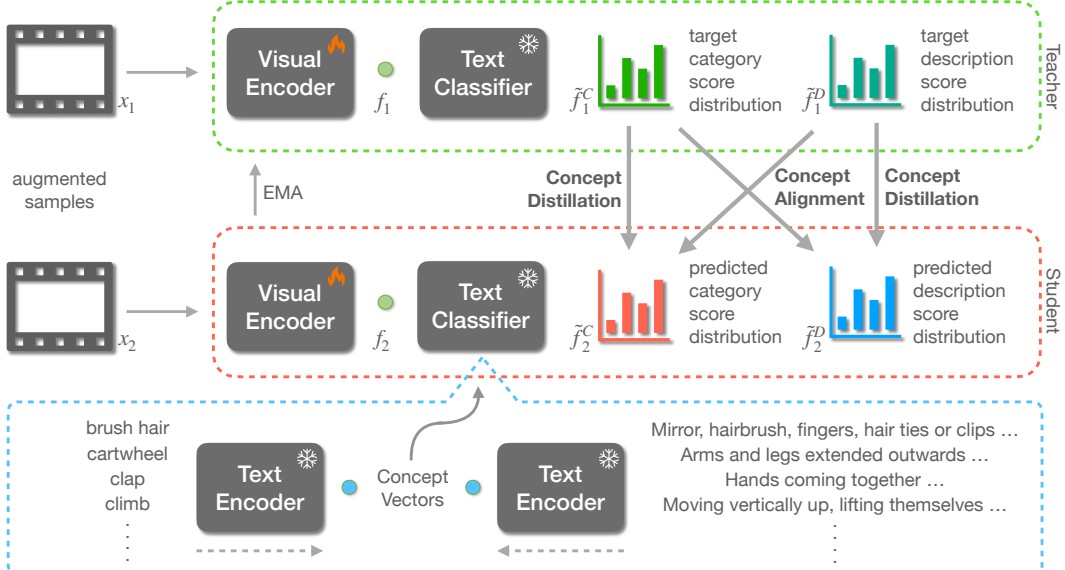

Figure 1: Our overall setup contains three components: visual teacher model (green), visual student model (red), and language model (blue). We utilize the text encoder of CLIP as our language model and extract *concept vectors* relevant to action labels and descriptions of those actions. A visual encoder (containing a space-time backbone) is partially initialized with the visual encoder of CLIP and used to obtain sample specific features. The generated concept vectors are used to project these features to a *concept space* where our proposed *concept distillation* and *concept alignment* losses are applied.

# 3 Language-based Self-Supervision (LSS)

In this section, we present our proposal, Language-based Self-Supervision (LSS). The generality and robustness of shared image-language representation spaces such as that of CLIP [4] allow interesting manipulations of visual representations using language. We explore such manipulations under the setting of visual self-supervised learning focusing on video understanding. Self-supervised objectives can operate within a latent space constructed with language, retaining language alignment of learned visual representations. This allows better interpretability of representations as well as zero-shot inference. We discuss the four key components of our approach: backbone architecture, concept distillation objective, modifications to avoid collapse, and concept alignment objective.

## 3.1 Backbone Architecture

Our approach introduces a *text classifier* to self-distillation based SSL works [15, 13], in place of the projector network. Given a data sample $x$, let $x_1, x_2 \in \mathbb{R}^{(C,T,H,W)}$ be two augmented views generated using video specific transformations following [13], where $C = 3, T = 8, H = W = 224$ are channel, time, and spatial dimensions respectively.

**Visual Encoder:** A visual encoder, $\theta_v$, processes $x_i$ to produce feature $f_i \in \mathbb{R}^{768}$. We utilize the pre-trained image encoder of CLIP [4] expanded for temporal modelling using factorized space-time attention. The vision transformer variant of CLIP is selected to allow our factorized space-time attention. In particular, we use ViT-B/16 architecture for the the image encoder, in which for a given augmented view with $H = W = 224$ and $T = 8$, each transformer block processes 8 temporal and 196 spatial tokens separately in sequential order, and the embedding dimension of each token is $\mathbb{R}^{768}$. In addition to the input tokens from the data sample, one classification token [64, 65] serves as the final feature vector output by the network, namely $f_i$, which is common to the CLIP image encoder. This classification token is inflated and processed suitably following [66] to accommodate our modifications for factorized space-time attention. We follow [66] to zero-initialize additional time-attention parameters, achieving outputs identical to the pre-trained CLIP image encoder at start of training.

**Text Classifier:** Inspired by [67], a set of $n$ language embeddings extracted from the CLIP text encoder, $\theta_t$, are used to construct the weight parameter of a linear layer (with no bias term), which

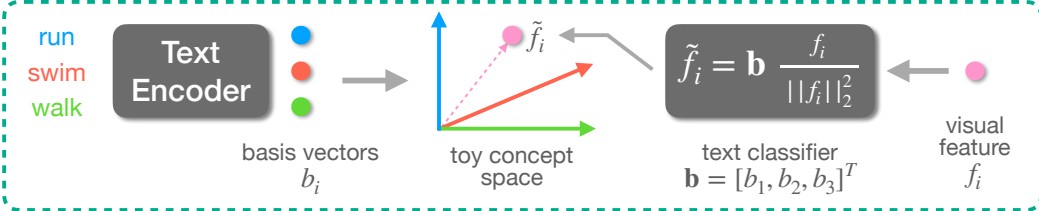

Figure 2: We illustrate a toy concept space constructed with the three action concepts: run, swim, and walk. In this example, the text classifier projects visual feature $f_i$ into the 3-dimensional toy concept space to produce $\tilde{f}_i$.

we call our text classifier, $\theta_c$. The role of this text classifier is to project visual features $f_i$ to a vector space defined by those $n$ embeddings, producing $\tilde{f}_i \in \mathbb{R}^n$. Next we discuss these vector spaces (referred to as action concept spaces) and the text classifier module in detail.

## 3.2 Action Concept Spaces

Self-supervised learning approaches following exponential moving average (EMA) based self-distillation [68, 15, 13] utilize a projector network (MLP) to operate in a higher dimensional feature space. This is expected to minimize train-test domain gaps, handle noisy positive sample pairs, and better discriminate nuanced feature differences [69]. Focused on these notions, we propose an alternate *concept space* composed of a set of basis vectors defined by language-based action concepts. Our language-based self-supervision objectives operate within such concept spaces.

**Concept Spaces:** Building off the assumption that text encoder features capture subtle differences between distinct actions categories, we hypothesize that necessary nuanced distinctions between these actions will be better captured in our proposed concept spaces. The defining parameters of concept spaces are their basis vectors, $b_i$. Normalized embeddings (extracted from text encoder, $\theta_t$) of various natural language captions ($c_i$) relevant to action categories are used as these basic vectors.

$$b_i = \theta_t(c_i) \ / \ ||\theta_t(c_i)||_2^2 \tag{1}$$

$$\mathbf{b} = [b_1, b_2, ... \ b_n]^T \ ; \ \mathbf{b} \in \mathbb{R}^{(n,d)} \tag{2}$$

Note that these basis vectors are not necessarily orthogonal. As illustrated in Fig. 2, a single set of basis vectors, $\mathbf{b}$, defines one action concept space. We define two sets of basic vectors: action category vectors and action description vectors. Action category vectors relate to a single action label which is converted to a caption using textual prompting following [4]. Action description vectors are averaged embeddings of multiple descriptions and visual characteristics relevant to individual action categories. These two distinct sets of basic vectors lead to two distinct concept spaces which we name *category concept space* and *description concept space* respectively.

**Category Concept Space:** We explore 3 different strategies to construct the category concept space. The base setup uses action labels from Kinetics-400 [70], UCF-101 [71], and HMDB-51 [72] datasets, leading to a set of 530 (400 + 101 + 51, ignoring overlaps) basis vectors. Our next goal of connecting LLMs and their action awareness occurs in the second two strategies. We utilize LLMs [18] and visual-LLMs [73] to extract large sets of action category labels. While we explore this idea of expanding the basis vector set with LMM based additional action labels in Section 4, the base setup containing a modest 530 categories was sufficient to improve downstream task performance.

**Description Concept Space:** This space is constructed conditioned on the previous category concept space. For each action label used in the latter, we extract 4 distinct descriptions and a set of visual characteristics relevant to that action label using a large language model (LLM). The role of the LLM is to inject its world knowledge (i.e. awareness on videos, actions, and their attributes) into our learned representations during self-supervised learning. In detail, we prompt GPT-3 [18] to generate such descriptions and characteristics using procedure outlined in Appendix A. We highlight that GPT-3 is used here as an intelligent LLM containing world knowledge on videos and actions, in order to create natural language descriptions for given action category labels. The textual outputs generated for each action label are processed by our text encoder to produce multiple embeddings for a single action label. These embeddings are averaged to produce the corresponding basis vector for the description concept space. Note how this leads to a common dimensionality between the two concept spaces as well as one to one correspondences between the basic vectors of the spaces, which we leverage in our self-supervision objectives.

## 3.3 Concept Distillation

We now describe our primary self-supervised learning objective, concept distillation. Standard multi-view based self-supervision enforces a network to encode the common information between two augmented (distorted) views of a data sample [69]. This common information can be considered as the augmentation invariant signal present in the original data sample [69, 74]. In the case of self-distillation based approaches [15, 13], a higher dimensional feature space is utilized to enforce the self-supervision objectives. Instead, we propose to use action concept spaces as an alternative.

Proposed concept distillation depends on an action concept space and visual video features aligned to the basis vectors of that space. Given our visual features $f_i \in \mathbb{R}^d$, we obtain projected $\tilde{f}_i \in \mathbb{R}^n$ as,

$$\tilde{f}_i = \mathbf{b}\,(\,f_i/\,||f_i||_2^2\,) = [b_1 \cdot f_i', b_2 \cdot f_i', ... b_n \cdot f_i']^T \tag{3}$$

**Similarity Calculation:** Projecting normalized visual video features to a concept space corresponds to calculating the dot-product similarity with each basic vector of the concept space. The projected vector $\tilde{f}_i$ can be viewed as a similarity *score distribution* across all basis vectors of the concept space. Inspired by [67], we implement this similarity calculation as a linear layer with weight matrix $\mathbf{b}$ and bias terms zero. We refer to this layer as the *text classifier*. Similar to [67], our text classifier remains frozen (no parameter updates), but in our case, this is to retain the original language distribution.

**Concept Distillation Objective:** Viewing projected features for two augmented views of a single video as score distributions, we argue that the underlying signal of the original video would relate to a unique score distribution to which score distributions of each view should be similar. Therein, following our EMA teacher based self-distillation setup (see Section 3.1 for details), we enforce the score distribution to be consistent across views. Given two views $x_1, x_2$ of a single video, our teacher and student visual encoders process them respectively to produce $f_1, f_2$. The text classifier projects these to concept space, producing score distributions $\tilde{f}_1, \tilde{f}_2$. We obtain our objective, $\mathcal{L}_{\text{CD}}$ as:

$$\hat{f}_i[k] = \frac{\exp(\tilde{f}_i[k]/\lambda_i)}{\sum_{j=1}^n \exp(\tilde{f}_i[j]/\lambda_i)} \tag{4}$$

$$w_s = \max(\hat{f}_1) \tag{5}$$

$$\mathcal{L}_{\text{CD}}(\tilde{f}_1, \tilde{f}_2) = -w_s \cdot \sum_{j=1}^n \hat{f}_1[j] \log \hat{f}_2[j] \tag{6}$$

The teacher and student score distributions, $\tilde{f}_1, \tilde{f}_2$, are softmax normalized in Eq. (4), with temperature terms $\lambda_1 = 0.1, \lambda_2 = 1$ for sharpening only the teacher score distribution. A significance score $w_s$ is calculated for each sample in Eq. (5). In the softmax normalized teacher score distribution ($\hat{f}_1$), the maximum value is high when peaked at a single action concept and low when peaked at multiple action concepts. Considering the noisy nature of multi-peak teacher score distributions, we utilize $w_s$ to minimize their overall effect during training. Our overall $\mathcal{L}_{\text{CD}}$ is thus implemented as in Eq. (6).

**Distinct Concept Spaces:** Given the two distinct action concept spaces defined in Section 3.2, we utilize two parallel text classifiers to implement each, and obtain two score distributions, one for each concept space. Defining score distributions $\tilde{f}_i^C, \tilde{f}_i^D$ for category and description concept spaces respectively, we apply our $\mathcal{L}_{\text{CD}}$ on each pair separately to obtain two losses $\mathcal{L}_{\text{CD}}^X$ for X∈{C,D} as:

$$\mathcal{L}_{\text{CD}}^X = \mathcal{L}_{\text{CD}}(\tilde{f}_1^X, \tilde{f}_2^X) \tag{7}$$

We highlight how our concept spaces implemented as text classifiers are maintained intact by freezing the text classifier during training. This allows our approach to perform direct zero-shot inference, making concept distillation additionally advantageous over standard video SSL techniques.

## 3.4 Uniform Distribution Prior

Avoiding collapse is a key concern in SSL methods [15, 13, 69] and recent self-distillation based approaches utilize feature sharpening and centering operations to avoid collapse [15, 13]. While we similarly perform sharpening operations on the teacher outputs, given the nature of our action concept space, performing a learned vector mean subtraction based centering operations can break the meaningful structure of score distributions. Instead, we enforce a uniform distribution prior on

the expected score distribution over the entire training dataset. The centering operation proposed in [15] acts similarly pushing representations towards a uniform distribution while the sharpening operation counters its effect. We approximate expectation over the dataset as a moving average of mean score distributions at each train iteration and the uniform prior is enforced as:

$$\hat{f}_{\text{MA}}^{\text{X}} = \tau \cdot \hat{f}_2^{\text{X}} + (1 - \tau) \cdot \hat{f}_{\text{MA}}^{\text{X}} \tag{8}$$

$$\mathcal{L}_{\text{UP}}^{\text{X}} = -\frac{1}{n} \sum_j \log \hat{f}_{\text{MA}}^{\text{X}}[j] \tag{9}$$

where the hyper-parameter $\tau = 0.5$ is fixed during training. We highlight that $\mathcal{L}_{\text{UP}}$ is necessary for convergence with concept distillation and is added to the concept distillation objective, $\mathcal{L}_{\text{CD}}^{\text{X}}$.

## 3.5 Concept Alignment

Aligning action category labels and their descriptions or attributes within some embedding space has been explored in video SSL under multiple settings [48, 49]. Motivated by these promising results, we explore how such alignment can be integrated to improve our framework with *concept spaces*. In Section 3.2, we define two distinct action concept spaces constructed from category labels and detailed category descriptions respectively. We hypothesize that explicit alignment of video features between these two spaces based on their one to one relationship can learn additional information. Therein, we introduce our concept alignment objective, $\mathcal{L}_{\text{CA}}$, as follows:

$$\mathcal{L}_{\text{CA}} = \mathcal{L}_{\text{CD}}(\tilde{f}_1^{\text{C}}, \tilde{f}_2^{\text{D}}) + \mathcal{L}_{\text{CD}}(\tilde{f}_1^{\text{D}}, \tilde{f}_2^{\text{C}}) \tag{10}$$

**Overall SSL Objective:** Reusing $\mathcal{L}_{\text{CD}}$ from Eq. (6), we match score distributions across our two concept spaces instead of within a single concept space. $\mathcal{L}_{\text{CD}}(\tilde{f}_1^{\text{C}}, \tilde{f}_2^{\text{D}})$ aligns student description score distribution $\tilde{f}_2^{\text{D}}$ to teacher category score distribution $\tilde{f}_1^{\text{C}}$ while $\mathcal{L}_{\text{CD}}(\tilde{f}_1^{\text{D}}, \tilde{f}_2^{\text{C}})$ aligns student category score distribution $\tilde{f}_2^{\text{C}}$ to teacher description score distribution $\tilde{f}_1^{\text{D}}$. Combining all terms, we obtain:

$$\mathcal{L} = (\mathcal{L}_{\text{CD}}^{\text{C}} + \mathcal{L}_{\text{UP}}^{\text{C}}) + (\mathcal{L}_{\text{CD}}^{\text{D}} + \mathcal{L}_{\text{UP}}^{\text{D}}) + \mathcal{L}_{\text{CA}} \tag{11}$$

## 3.6 Concept Space Variants

Our baseline concept space (described in Section 3.2) utilizes labels from three standard video datasets (Kinetics-400, UCF-101, HMDB-51). However, we want to ensure scalability with more data and no label leakage to downstream evaluation tasks. With this goal, we propose 2 additional variants of action concept spaces tagged LSS-B and LSS-C. These variants do not use any form of ground truth textual labels from datasets. Moreover, they leverage the world awareness (i.e. knowledge on videos and actions) of LLMs to generate extensive action categories. Our baseline setup is hereafter referred as LSS-A.

For LSS-B, we use GPT-3 [18] to generate a large set of action labels. We first prompt GPT to categorize all common human actions / activities into 20 groups. For each group, we again ask GPT to generate at least 100 visually diverse action categories. These are all collected to create a set of 2000 action labels. We then use projections of these labels in CLIP text-encoder representation space to eliminate labels of high semantic similarity (spectral clustering in feature space from [75] to identify similar features), achieving 1000 diverse action categories. So our 1000 action categories for LSS-B are generic, not tied to any of our training datasets, and scalable with more data.

For LSS-C, we generate a label set using only videos from the training dataset. We use PCA based clustering to identify 2000 representative videos from a randomly sampled subset (50,000) of our training dataset and then use image-captioning models (LLaVa [73]) on video center frames to generate a diverse set of 2000 action labels. This is further reduced to 500 eliminating labels that are similar in feature space of the CLIP text encoder. In this case, our generated labels are tied to the training dataset, but uses no textually annotated category labels. We use only the videos (and an image-to-text captioning model) to generate our label set, still resulting in a scalable framework.

Note that each of these alternate strategies relates to construction of our category concept space. Given the selected set of textual category labels of this space, the description concept space is constructed in the same common way (as described in Section 3.2). We also reiterate that LSS-B and LSS-C variants use no category information from train / test datasets.

# 4 Experiments

In this section, we first describe our experimental setup followed by discussion of results for linear probing self-supervised representations and zero-shot analysis.

**Datasets:** We use three standard action recognition benchmark datasets in our experiments: Kinetics-400 [70], UCF-101 [71], and HMBD-51 [72]. Kinetics-400 is a large-scale dataset containing 240,000 training videos and 20,000 validation videos belonging to 400 different action classes. On average, these videos are of duration around 10 seconds, with 25 frames per second (i.e., around 250 frames per video). UCF-101 and HMBD-51 are small-scale datasets each containing 13k videos (9.5k/3.7k train/test) belonging to 101 classes and 5k (3.5k/1.5k train/test) videos belonging to 51 classes respectively. They also contain similar duration videos.

**Self-supervised Training:** Our SSL training phase uses the train split of Kinetics-400 dataset [70] *without* using any per-video labels. We train for 15 epochs using a batch size of 32 across 4 NVIDIA-A5000 GPUs using ADAM-W [76, 77] optimizer on the student model with an initial learning rate of $1e-5$ following a cosine decay schedule. The EMA teacher is updated from student weights after each training iteration with a decay ratio of $2e-4$. Unless otherwise specified, this model is used for all downstream task evaluations.

**Transductive Training:** For selected experiments, we additionally perform self-supervised training directly on the train split of each downstream dataset. For Kinetics-400, we follow the same setup described above. In the case of HMDB-51 and UCF-101, we perform self-supervised training for a longer duration of 100 epochs (smaller train sets) leaving all other hyper-parameters unchanged.

**View Generation:** Our self-supervised setup requires two views of a single video. We sample two clips from a video following global view generation in [13]. In detail, we select two random intervals from a video, and uniformly sample (equal time gaps between frames) 8 frames of 224x224 spatial dimensions from within that interval. Standard video augmentations from [78] are also applied randomly for each view.

**Linear Probing:** We follow standard linear probing settings on our two downstream datasets to evaluate quality of representations learned by our self-supervised learning phase. We follow the same settings in [13] for fair comparison. Our visual encoder is frozen and a randomly initialized linear layer is trained on the train split of the downstream dataset in a fully-supervised manner. We train for 15 epochs using a batch size of 128 across 4 NVIDIA-A5000 GPUs using ADAM-W [76, 77] optimizer with an initial learning rate of $1e-3$ following a cosine decay schedule. During inference, we sample three 224x224 dimensional spatial crops with 8 uniformly spaced frames from each video following prior work [13, 36].

**Zero-Shot Inference:** For zero-shot inference, we project class labels of downstream datasets to our text encoder feature space, and construct an alternate text classifier. Using this text classifier, we make zero-shot predictions. This setup is identical to dot-product similarity based inference in CLIP [4] (explanation in Section 3.3). In line with prior work [13, 36], we feed three 224x224 dimensional spatial crops with 8 uniformly spaced frames sampled from each video to the visual encoder and average its output feature embedding prior to normalized dot-product calculation in the text encoder.

## 4.1 Linear-Probing Analysis

We first evaluate LSS under linear probing settings on HMDB-51 & UCF-101 datasets. Our results (top-1 accuracy) are reported in Table 1. Our proposed LSS achieves state-of-the-art results on both datasets, outperforming prior approaches. Note that MoDist [81] and BraVe [89], both of which additionally utilize video-level optical flow (OF) for self-supervision, are not directly comparable. Still, our LSS showcases competitive performance to those, even without such motion information.

## 4.2 Zero-Shot Analysis

Our LSS provides the additional advantage of zero-shot operation unlike standard video SSL approaches. To this end, we conduct two forms of zero-shot experiments. First, we evaluate LSS on standard zero-shot classification, where our model trained on Kinetics-400 (under SSL settings) is evaluated on the two downstream datasets, HMDB-51 and UCF-101. We report these results (top-1 accuracy) in Table 2. Compared to prior work utilizing per-video labels / captions for training, we

Table 1: **Linear Probing on HMDB-51 [72] and UCF-101 [79]:** We compare our method against prior work, reporting top-1 (%) accuracy (following evaluation procedure in [13]). 'ITP' stands for image-text pre-training. Gray shaded methods use additional optical flow (OF) inputs for training. Nevertheless, our performance is comparable to such methods using per-video OF modality information. In contrast, we use generic language modality information and require no one-to-one language relations with individual videos during training.

| Method | Backbone | ITP | TFLOPS | Frames | Epochs | HMDB | UCF |
|---|---|---|---|---|---|---|---|
| MemDPC [34] (ECCV '20) | R2D3D-34 | ✗ | - | 64 | - | 30.5 | 54.1 |
| CoCLR [35] (NeurIPS '20) | S3D | ✗ | 0.07 | 32 | 100 | 52.4 | 77.8 |
| VideoMoCo [80] (CVPR '21) | R(2+1)D | ✗ | 17.5 | 32 | 200 | 49.2 | 78.7 |
| CVRL [36] (CVPR '21) | R3D-50 | ✗ | 3.19 | 32 | 800 | 57.3 | 89.2 |
| MoDist [81] (Arxiv '21) | R3D-50 | ✗ | 3.19 | 8 | 100 | 63.0 | 91.5 |
| BraVe [38] (ICCV '21) | R3D-50 | ✗ | 3.19 | 16 | - | 68.3 | 92.5 |
| Vi$^2$CLR [82] (ICCV '21) | S3D | ✗ | 0.07 | 32 | 300 | 47.3 | 75.4 |
| MCN [83] (ICCV '21) | R3D | ✗ | 3.19 | 32 | 50 | 42.9 | 73.1 |
| CORP [84] (ICCV '21) | R3D-50 | ✗ | 3.19 | 16 | 800 | 58.7 | 90.2 |
| SVT [13] (CVPR '22) | ViT-B | ✗ | 0.59 | 16 | 20 | 57.8 | 90.8 |
| VideoMAE [14] (NeurIPS '22) | ViT-B | ✗ | 0.59 | 16 | 800 | 60.3 | 84.7 |
| MERLOT [85] (NeurIPS '21) | ViT-B | ✓ | - | 16 | - | 55.4 | 80.1 |
| VATT [86] (NeurIPS '21) | ViT-B | ✓ | - | 32 | - | 66.4 | 87.6 |
| TVTS [87] (CVPR '23) | ViT-B | ✓ | 0.59 | 16 | 20 | 58.4 | 83.4 |
| LaViLa [88] (CVPR '23) | ViT-L | ✓ | - | 4 | 5 | 61.5 | 88.1 |
| LSS-A (ours) | ViT-B | ✓ | 0.59 | 8 | 20 | 69.2 | 91.0 |
| LSS-B (ours) | ViT-B | ✓ | 0.59 | 8 | 20 | **69.4** | **91.1** |
| LSS-C (ours) | ViT-B | ✓ | 0.59 | 8 | 20 | 69.1 | 90.8 |

Table 2: **Zero-shot Transfer on HMDB-51 [72] and UCF-101 [79]:** We compare LSS against prior work, reporting top-1 accuracy (%). Mean across three test splits is reported following [55]. 'ITP' stands for image-text pre-training and 'Video Labels' refers to using per-video annotations (or paired captions) for supervision during video-based training. We highlight how among directly comparable unsupervised (at video level) approaches as well as over the CLIP [4] baseline, LSS boosts zero-shot performance.

| Method | Backbone | ITP | Video Labels | Frames | HMDB | UCF |
|---|---|---|---|---|---|---|
| TS-GCN [90] (AAAI '19) | GCN | ✗ | ✓ | 16 | 23.2 | 34.2 |
| E2E [91] (CVPR '20) | CNN | ✗ | ✓ | 16 | 32.7 | 48.0 |
| ER-ZSAR [48] (ICCV '21) | CNN | ✗ | ✓ | 8 | 35.3 | 51.8 |
| ActionCLIP [11] | ViT-B | ✓ | ✓ | 32 | 40.8 | 58.3 |
| X-CLIP [12] (ECCV '22) | ViT-B | ✓ | ✓ | 32 | 44.6 | 72.0 |
| VicTR [55] | ViT-B | ✓ | ✓ | 32 | 51.0 | 72.4 |
| ViFi [62] (CVPR '23) | ViT-B | ✓ | ✓ | 32 | 51.3 | 76.8 |
| MTE [92] (ECCV '16) | - | ✗ | ✗ | - | 19.7 | 15.8 |
| ASR [93] (ECML '17) | CNN | ✗ | ✗ | 16 | 21.8 | 24.4 |
| ZSECOC [94] (CVPR '17) | - | ✗ | ✗ | - | 22.6 | 15.1 |
| UR [49] (CVPR '18) | CNN | ✗ | ✗ | 1 | 24.4 | 17.5 |
| CLIP [4] (ICML '21) | ViT-B | ✓ | ✗ | 1 | 46.5 | 69.8 |
| CLIP [4] (ICML '21) | ViT-B | ✓ | ✗ | 8 | 47.2 | 70.3 |
| LaViLa [88] (CVPR '23) | ViT-L | ✓ | ✗ | 4 | 16.6 | 18.2 |
| LSS-A (ours) | ViT-B | ✓ | ✗ | 8 | 49.5 | 72.0 |
| LSS-B (ours) | ViT-B | ✓ | ✗ | 8 | 50.2 | 73.8 |
| LSS-C (ours) | ViT-B | ✓ | ✗ | 8 | **51.4** | **74.2** |

achieve competitive performance. We note that MOV [7] trained under supervised settings with per-video labels and additional audio information is not a direct comparison.

In contrast to most prior approaches, LSS uses no video level labels for its Kinetics-400 training. In particular, LSS has not seen any labelled videos during its training process. Compared to prior

Table 3: **Transductive Zero-shot Transfer on HMDB-51 [72], UCF-101 [79], and Kinetics-400 [70]:** We report top-1 accuracy (%) following the evaluation procedure in [55]. Similar to prior work, we perform dataset specific unsupervised fine-tuning (using our self-supervised objective) on the train-splits of each downstream dataset (no labels used). 'ITP' refers to image-text pretaining, and 'Video Labels' refers to video level supervised training. Note that CLIP [4] is not transductive and is included only for comparison purposes.

| Method | Backbone | ITP | Video Labels | Frames | HMDB | UCF | K400 |
|---|---|---|---|---|---|---|---|
| UR [49](CVPR '18) | CNN | ✗ | ✗ | 1 | 28.9 | 20.1 | - |
| TS-GCN [90](AAAI '19) | GCN | ✗ | ✓ | 16 | 23.2 | 34.2 | - |
| CLIP [4](ICML '21) | ViT-B | ✓ | ✗ | 8 | 47.2 | 70.3 | 49.7 |
| MUST [59](ICLR '23) | ViT-B | ✓ | ✗ | 1 | 48.9 | **81.1** | 51.2 |
| LSS (ours) | ViT-B | ✓ | ✗ | 8 | **55.0** | 75.6 | **54.3** |

Table 4: **Ablation on SSL objectives:** We ablate our proposed concept distillation (CD) applied on category ($CD^C$) and description ($CD^D$) concept spaces and concept alignment (CA) using linear probing (LP) & zero-shot transfer (ZS) on HMDB-51 [72] dataset. Since our approach cannot be trained without any objective, we construct two new baselines from CLIP [4] and SVT [13] (details in Section 4.3). Each proposed component obtains clear improvements over the baselines.

| | $CD^C$ | $CD^D$ | CA | LP | ZS |
|---|---|---|---|---|---|
| CLIP | ✗ | ✗ | ✗ | 63.9 | 46.5 |
| CLIP$^\dagger$ | ✗ | ✗ | ✗ | 67.3 | 47.2 |
| SVT$^\S$ | ✗ | ✗ | ✗ | 62.2 | - |
| Ours | ✓ | ✗ | ✗ | 68.5 | 48.8 |
| Ours | ✓ | ✓ | ✗ | 69.0 | 49.2 |
| Ours | ✓ | ✓ | ✓ | 69.2 | 49.5 |

Table 5: **Concept Space Ablation:** We report zero-shot accuracy (%) on HMDB-51 and UCF-101 datasets. 'K400 only' shows transfer to unseen downstream classes. The labels of K400 overlapping with UCF/HMDB are not used here. A CLIP [4] baseline (modified for video domain without re-training) is reported in row 1 for comparison purposes. For 2000 words & 10,000 sentences, we utilize most common nouns / verbs and example sentences for action verbs from WordNet [95, 96].

| Method | Action Labels | HMDB | UCF |
|---|---|---|---|
| CLIP | - | 47.2 | 70.3 |
| Ours | K400 only (w/o U,H) | 48.4 | 71.1 |
| Ours | K400+U+H (A) | 49.5 | 72.0 |
| Ours | (A)+2000 words | 48.6 | 71.8 |
| Ours | (A)+10K sentences | 49.6 | 71.4 |

work operating under these settings, LSS achieves state-of-the-art performance on both downstream datasets as seen in the bottom half of Table 2.

An alternate setting in prior zero-shot work is transductive training, where self-supervised learning is perfomed directly on train splits of downstream datasets. Under this setting, we evaluate on all three datasets, Kinetics-400, HMDB-51, and UCF-101, reporting results (top-1 accuracy) in Table 3. In the case of HMBD-51 and Kinetics-400, our method achieves state-of-the-art performance. For UCF-101, we achieve competitive results, and clear improvements over a CLIP [4] baseline.

## 4.3 Ablations

We next study the contribution of each component within our approach. All ablative experiments follow the same SSL phase on the Kinetics-400 train set (as described in Section 4) followed by zero-shot analysis on validation sets of HMDB-51 and UCF-101. In the case of linear probing results, training is conducted following same settings (see Section 4) on the train set of HMDB-51 followed by evaluation on its validation set.

**SSL Objectives:** First we ablate each proposed component in Eq. (11) and report results in Table 4. In addition to a direct CLIP [4] baseline, we construct two additional baselines building off CLIP [4] and SVT [13] for better comparison. CLIP$^\dagger$ baseline applies our backbone modifications (for temporal modeling) with no training, which is identical to averaging per-frame visual encoder features. SVT$^\S$ baseline performs SVT [13] training with CLIP visual encoder initialization (note that language alignment breaks and zero-shot operation is not possible for this baseline). In comparison to the CLIP baselines, each proposed component, concept distillation in category and description concept spaces as well as concept alignment, leads to improvements. The comparison against SVT$^\S$ highlights how our SSL approach better preserves language aligned information (contained in CLIP) that is useful even in linear probing. In contrast, the lower performance of SVT$^\S$ compared to CLIP baselines indicates that generic SSL techniques may be losing useful information contained in CLIP.

Table 6: **Ablation on Regularization and Significance Weight:** The effect of proposed regularization (left) and significance weight (right) is demonstrated. UDP regularization (see Section 3.4) is particularly essential to prevent collapse during the SSL training phase. Significance weight ($w_s$) also improves performance.

| Method | HMDB | UCF |
|---|---|---|
| LSS | 48.4 | 71.1 |
| LSS w/o UDP | 33.4 | 54.3 |

| Method | HMDB | UCF |
|---|---|---|
| LSS | 48.4 | 71.1 |
| LSS w/o $w_s$ | 47.2 | 70.3 |

**Concept Spaces:** Our next focus is on construction of concept spaces. We explore how separately augmenting each concept space affects downstream task performance measured with zero-shot transfer. These results are reported in Table 5. First, focused on the category concept space, we construct additional category labels using 1000 most common nouns and verbs each (total of 2000) from the WordNet dataset [95, 96]. Next, we augment the description concept space using 10,000 sentences. We select these from example sentences provided for action verbs in the WordNet dataset [95, 96]. In these experiments, only the concept distillation objective is applied on these augmented spaces and concept alignment operates only on the base category set. This is because independently augmenting one of the action spaces eliminates their shared and aligned dimensionality. Results for these two settings are reported in row 2 & 3 respectively in Table 5.

**Uniform Distribution Prior:** We ablate on proposed uniform distribution prior (UDP) which acts as a regularization to prevent collapse (Section 3.4). Our results in Table 6 (left) indicate clear necessity of such regularization to prevent collapse during SSL training.

**Significance Weight in Concept Distillation:** In Eq. (6), we utilize a significance weight term, $w_s$, which represents the confidence of the target concept space projection for a given sample. We note how each sample during training is a clip sampled from a video (which covers a temporal crop of video). Our intuition for this weight is to act as a way of prioritizing more important clips over the less important ones. Our ablations in Table 6 (right) indicate usefulness of this weight term.

## 5 Conclusion

We introduce a novel language-based self-supervised learning (SSL) approach for videos, termed LSS, capable of adapting strong language-aligned image representations (CLIP [4]) to the video domain. In particular, we propose two self-distillation based SSL objectives, *concept distillation* and *concept alignment*. Our approach trains with no video level labels or paired captions similar to prior video SSL works, but retains language alignment from image CLIP enabling direct zero-shot inference. We demonstrate state-of-the art performance in terms of linear probing with the learned representations on downstream tasks. For zero-shot operation, LSS demonstrates strong performance under both standard and transductive settings, indicating a promising direction for video SSL.

**Limitations, Future Work, & Broader Impact**: The language alignment of LSS may be limited mostly to per-frame static information since the alignment is derived from image CLIP [4]. LSS cannot distinguish motion based categories like "moving object left to right". Moreover, while containing highly discriminative and generic information at image level, CLIP features lack spatial awareness at an object level [75]. Our proposed model building off these representations in inherently limited in understanding object level motion and interaction within videos. However, recent progress in localization aware CLIP models [75, 97, 98] opens avenues for leveraging their object-centric or pixel-level representations to better model such video motion patterns, opening up interesting future directions. In terms of broader impact, the datasets and pre-trained models we use possibly contain biases, which may be reflected in LSS. However, our reduced reliance on human annotations may lower additional biases.

**Reproducibility Statement**: We build a codebase derived from source code of SVT [13] & CLIP [4] and use pre-trained CLIP weights from `https://github.com/openai`. All experiments use publicly available datasets. Our action descriptions will be released publicly along with our codebase.

**Acknowledgements**: We thank Xiang Li for helpful discussions and server setup. We also thank Kumara Kahatapitiya and Cristina Mata for helpful discussions.

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

# APPENDIX

## A Prompting details

Our proposed approach utilizes two sets of language based captions: categories and descriptions. While categories are obtained directly from the class labels of datasets (set of unique labels - e.g. 400 classes in Kinetics-400 dataset), the descriptions are generated automatically utilizing GPT-3 [18]. For each category caption, we query GPT-3 to provide a set of descriptions and visual characteristics. In detail, we use the following two prompts to generate descriptions and visual characteristics:
```
prompt1 = "Give 4 different descriptions for the phrase: {category}?"
prompt2 = "List visual objects or characteristics usually seen with the
action: {category}?"
```
The resulting two sets of captions are converted to text embeddings using our text-encoder, and a single average text embedding is computed. This averaged embedding is used as the description basis vector for that category. Also, the resulting dataset containing these category-description pairs is made available publicly.

## B Additional Experiments

**Linear Probing Evaluation:** We present more results for linear probing in Table 7 (left). Our proposed LSS improves over the baseline achieving competitive performance on Kinetics-400.

**Text-to-video retrieval:** An important characteristic of CLIP [4] is its retrieval ability across both language and visual modalities. In order to verify if proposed LSS retains these strengths, we run experiments on MSR-VTT text-to-video retrieval benchmark. We demonstrate how LSS improves over our baseline CLIP, reporting these results in Table 7 (top-right).

**Charades Evaluation:** We explore an alternate task of zero-shot multi-label classification on the Charades video dataset. We report mAP results for this task in Table 7 (bottom-right) as an additional point of comparison.

Table 7: We report top-1 (%) accuracy on the Kinetics-400 [70] validation set for linear probing evaluation (left). All models are pre-trained on the training set of Kinetics-400 dataset. We also report a CLIP baseline for comparison purposes. Performance of our proposed approach is on-par with prior state-of-the-art and showcases improvements over our baseline method. We also report retrieval scores (top-right) for MSR-VTT and classification mAP (bottom-right) for Charades dataset.

| Method | Backbone | Acc (%) |
|---|---|---|
| CVRL [36] (CVPR'21) | R3D-101 | 67.6 |
| BraVe [38] (ICCV'21) | R3D-50 | 66.7 |
| Vi$^2$CLR [82] (ICCV '21) | S3D | 63.4 |
| CORP [84] (ICCV '21) | R3D-50 | 66.6 |
| SVT [13] (CVPR '22) | ViT-B | 68.1 |
| VideoMAE [14] (NeurIPS '22) | ViT-B | 61.3 |
| CLIP [4] | ViT-B | 66.4 |
| LSS (ours) | ViT-B | 67.3 |

| Method | R@1 | R@5 | R@10 |
|---|---|---|---|
| CLIP [4] | 30.6 | 54.4 | 64.3 |
| LSS (ours) | 33.8 | 58.2 | 70.3 |

| Method | Classification mAP |
|---|---|
| CLIP [4] | 19.7 |
| LSS (ours) | 23.1 |

