# OpenReview forum: "Language-based Action Concept Spaces Improve Video Self-Supervised Learning"
_NeurIPS.cc/2023/Conference — NeurIPS 2023 poster_

### Official Review · Reviewer_vhEc · 2023-06-29

**Soundness:** 3 good
**Presentation:** 3 good
**Contribution:** 2 fair
**Rating:** 5
**Confidence:** 4

**Summary:**

This paper proposes to transfer CLIP to the video domain for self-supervised learning. Textual features of video categories are used to obtain text classifiers and fixed during pre-training to obtain transferable information. Multiple complementary loss functions are designed for pre-training. Experimental results on three datasets demonstrate the effectiveness of the method.

**Strengths:**

Pros:
1. A new paradigm is proposed to transfer the knowledge of CLIP to the video domain for self-supervised learning.
2. Impressive performance has been achieved on multiple datasets.
3. The results of the ablation experiments demonstrate the effectiveness of the method.

**Weaknesses:**

Cons:
1. The authors claim that they did not use labeled or captioned videos in the paper. But they used the labels of the Kinetics-400, UCF101, and HMBD51 data sets during training. Does this violate the self-supervised setting?
2. Unfair comparison. The proposed LSS leverages the pretrained CLIP for weight initialization while existing methods such as SVT，VideoMAE do not.
3. LSS is pre-trained on the Kinetics dataset. Why did the label of HMBD51 and UCF101 be added to the pre-training? Does this break the downstream transfer setup? Since the target label has been leaked in the pre-training.
4. The fully-finetuned experiments and SSv2 results are missing.

**Questions:**

Please refer to Weaknesses for more details.

**Limitations:**

Limitations have been listed.

---

> ### Author Rebuttal · Authors · 2023-08-09
>
> We thank the reviewer for the positive comments and address all concerns below.
>
> 1. `Using dataset labels for training:` We understand this shortcoming and run two new experiments that use no textual labels from datasets for our action concept spaces. We report these results in main rebuttal PDF (LSS-B & C in Tables 1, 2) and highlight the on-par performance of these additional experiments. We also include these results for linear probing (top) and zero-shot (bottom) top-1 accuracy below for quick reference.
>
>     |    Method    | ITP | HMDB-51 |  UCF-101 |
>     |:------------:|:---:|:----:|:----:|
>     | LSS-A (ours) | yes | 69.2 | 91.0 |
>     | LSS-B (ours) | yes | **69.4** | **91.1** |
>     | LSS-C (ours) | yes | 69.1 | 90.8 |
>
>     | Method | Action Labels    | HMDB | UCF  |
>     |--------|------------------|------|------|
>     | CLIP   | -                | 47.2 | 70.3 |
>     | Ours   | K400+U+H (LSS-A)     | 49.5 | 72.0 |
>     | Ours   | GPT labels (LSS-B)   | 50.2 | 73.8 |
>     | Ours   | I-VLM labels (LSS-C) | **51.4** | **74.2** |
>
>     This highlights how LSS can operate without using dataset labels for training.
>
>
> 2. `Unfair comparison`: We have updated Table 1 to compare against methods using CLIP pre-trained weights for initialization (reported in main rebuttal PDF). We confirm that LSS also outperforms these prior works taking advantage of CLIP pre-training.
> We also highlight the zero-shot abilities inherent to our LSS different from traditional SSL approaches.
>
>
> 3. `Adding UCF & HMDB to pre-training action set:` We thank the reviewer for this suggestion. Our experiments with LSS-B and LSS-C confirms that adding downstream dataset action classes is not necessary for strong performance. We will highlight this more in the final version of our paper. The motivation behind adding these UCF & HMDB labels in our initial experiment was to confirm capabilities of LSS given all possible action categories. Our further experiments as suggested above illustrate that even without such downstream labels, LSS achieves strong performance better than prior work. These results are presented in Table 2 (rebuttal PDF). We repeat these results below for quick reference.
>
>     | Method | Action Labels | HMDB | UCF  |
>     |--------|---------------|------|------|
>     | CLIP   | -             | 47.2 | 70.3 |
>     | Ours   | K400+U+H      | 49.5 | 72.0 |
>     | Ours   | K400 only     | 48.4 | 71.1 |
>     | Ours   | GPT labels (LSS-B)   | 50.2 | 73.8 |
>
>
> 4. `Experiment on more datasets:`
> We report mAP results for *zero-shot multi-label classification* task on Charades video dataset below as an additional point of comparison:
>
>     | Method |  Charades mAP |
>     |:------:|:----:|
>     |  CLIP  | 19.7 |
>     |  LSS-B | 23.1 |
>
>     In the case of motion heavy datasets like SSv2, we note that LSS poses limitations (since it has no language awareness for motion). We hope to explore this direction further in future work.

---

> > ### Comment · Reviewer_vhEc · 2023-08-16
> >
> > Thanks to the authors' responses, some of my concerns were addressed. But I still question whether the task setting itself is Self-supervised, because it needs to get the category annotation of the action. Although the author also carried out the experiment of GPT labels, in fact, the production of GPT-label also requires a real action label. Based on the above observations and the comments of other reviewers, I decided to keep the original score, and hope that the author can modify and improve the paper according to the comments of reviewers.

---

> > > ### Author Response · Authors · 2023-08-16
> > > **LSS-B & C require no annotated action labels**
> > >
> > > We thank the reviewer for their comments, but we would like to clarify that,
> > > 1. Experiments with LSS-B & LSS-C **require no annotated action labels** for learning. They use the same data (only videos) as traditional SSL methods during the self-supervised learning phase.
> > > 2. These variants obtain on-par (or better) performance to using annotated action labels

---

> > > > ### Comment · Reviewer_vhEc · 2023-08-16
> > > >
> > > > Thanks to the authors for the quick reply. In the newly uploaded PDF file, the prompt of the GPT label is 'prompt1 = "Give 4 different descriptions for the phrase: {category}?"', isn't the {category} here the real action label? Please correct me if my understanding is wrong, thanks.

---

> > > > > ### Author Response · Authors · 2023-08-16
> > > > >
> > > > > We apologize for the lack of clarity on our part.
> > > > >
> > > > > For LSS-B, we use GPT to generate a large set of action labels. We first prompt GPT to categorize all common human actions / activities into 20 groups. For each group, we again ask GPT to generate at least 100 visually diverse action categories. These are all collected to create a set of 2000 action labels. We then use projections of these labels in CLIP text-encoder representation space to eliminate labels of high semantic similarity, achieving only 1000 diverse action categories. So our 1000 action categories for LSS-B are generic, and not tied to any of our training datasets.
> > > > >
> > > > > For LSS-C, we generate a label set using only videos from the training dataset. We use PCA based clustering to identify 2000 representative videos from a randomly sampled subset (50,000) of our training dataset and then use image-captioning models on video center frames to generate a diverse set of 2000 action labels. This is further reduced to 500 eliminating labels that are similar in feature space of the CLIP text encoder. In this case, our generated labels are tied to the training dataset, but uses no annotated category labels. We use only the videos (and an image-to-text captioning model) to generate our labels set.
> > > > >
> > > > > These generated label sets are then used as the category labels (in place of the real category labels). They are treated as real category labels for the rest of our SSL training. Also, in response to:
> > > > > > 'prompt1 = "Give 4 different descriptions for the phrase: {category}?"', isn't the {category} here the real action label?
> > > > >
> > > > > *'category'* here is our generated action label (not using any annotation to generate). It is not the real action label.
> > > > >
> > > > > We will ensure to highlight these details better in our revised manuscript.

---

> > > > > > ### Comment · Reviewer_vhEc · 2023-08-16
> > > > > >
> > > > > > Thanks to the authors' responses, some of my concerns were addressed. The advantages of this paper, such as the use of text information to guide representation learning, and disadvantages, such as the proposed method lacks a strong baseline as a support and may have poor performance on SSv2 dataset.  Based on the above analysis, I decided to change my score from reject to borderline reject, and I will give my final score based on the discussions/comments with other reviewers.

---

> > > > > > > ### Author Response · Authors · 2023-08-16
> > > > > > >
> > > > > > > We thank the reviewer for their consideration and valuable feedback.

---

> > > > > > > > ### Author Response · Authors · 2023-08-16
> > > > > > > > **Comparison to Strong Baselines**
> > > > > > > >
> > > > > > > > With regard to strong baselines, we kindly highlight the multiple strong baselines that use **same image-text pretraining (ITP)** settings as our work, that we reported in the rebuttal. We outperform all of these approaches. LSS even outperforms the recent TVTS and LaViLa published at CVPR'23.
> > > > > > > >
> > > > > > > > We reiterate the results for linear probing below (top-1 accuracy %).
> > > > > > > >   |    Method    | ITP | HMDB-51 |  UCF-101 |
> > > > > > > >   |:------------:|:---:|:----:|:----:|
> > > > > > > >   |    MERLOT    | yes | 55.4 | 80.1 |
> > > > > > > >   |     VATT     | yes | 66.4 | 87.6 |
> > > > > > > >   |     TVTS     | yes | 58.4 | 83.4 |
> > > > > > > >   |    LaViLa    | yes | 61.5 | 88.1 |
> > > > > > > >   | LSS-A (ours) | yes | 69.2 | 91.0 |
> > > > > > > >   | LSS-B (ours) | yes | **69.4** | **91.1** |
> > > > > > > >   | LSS-C (ours) | yes | 69.1 | 90.8 |
> > > > > > > >
> > > > > > > > We also reiterate the results for zero-shot classification below (top-1 accuracy %).  Methods tagged * are fully-supervised.
> > > > > > > >
> > > > > > > >   | Method                    | ITP | HMDB | UCF  |
> > > > > > > >   |---------------------------|-----|------|------|
> > > > > > > >   | CLIP                      | yes | 46.5 | 69.8 |
> > > > > > > >   | CLIP (modified for video) | yes | 47.2 | 70.3 |
> > > > > > > >   | ActionCLIP *                   | yes | 40.8 | 58.3 |
> > > > > > > >   | X-CLIP *                   | yes | 44.6 | 72.0 |
> > > > > > > >   | VicTR *                   | yes | 51.0 | 72.4 |
> > > > > > > >   | LaViLa                    | yes | 16.6 | 18.2 |
> > > > > > > >   | LSS-A (ours)              | yes | 49.5 | 72.0 |
> > > > > > > >   | LSS-B (ours)              | yes | 50.2 | 73.8 |
> > > > > > > >   | LSS-C (ours)              | yes | **51.4** | **74.2** |
> > > > > > > >
> > > > > > > > While noting how some comparisons here are very recent (e.g. CVPR'23), we apologize for any lack of clarity on our part in adequately highlighting these results. We will clearly include these strong baselines in our main result tables in the revised final manuscript. We kindly ask the reviewer if they are able to take into consideration these stronger baselines.
> > > > > > > >
> > > > > > > > Further, we would be happy to provide comparisons to any additional strong baselines suggested by the reviewer.

---

> > > > > > > > > ### Comment · Reviewer_vhEc · 2023-08-18
> > > > > > > > >
> > > > > > > > > Thanks to the authors for their detailed responses. Some of my concerns are solved. I decide to give a borderline accept score, and I will also give my final score based on the discussions with other reviewers.

---

> > > > > > > > > > ### Author Response · Authors · 2023-08-20
> > > > > > > > > >
> > > > > > > > > > We thank the reviewer again for taking into consideration our results from the rebuttal. We will update our final draft with all material from the rebuttal.

---

### Official Review · Reviewer_aWeV · 2023-07-02

**Soundness:** 3 good
**Presentation:** 2 fair
**Contribution:** 3 good
**Rating:** 6
**Confidence:** 4

**Summary:**

The paper proposes a new self-supervised approach to adapt image-level CLIP features to video. The key idea is to use a teacher-student self-supervised learning framework, and distill the knowledge in the action concept space, derived from text action concepts using the CLIP's text encoder. The resulting framework produces SoTA self-supervised video features.

**Strengths:**

- The work tackles an important problem - adapting the CLIP visual encoder to video. Doing it without supervision is nice a bonus here.
- The approach is simple and sound.
- The evaluation is extensive and the results are strong.

**Weaknesses:**

- Since the set of concept vectors is taken from Kinetics-400  UCF-101 and HMDB-51, where the evaluation is performed, it is a little unfair to call the method completely unsupervised. While it is true that no video labels are used during training, the training process does use the knowledge of the actions vocabulary, which would  aid the evaluation. To avoid this shortcoming, the authors may consider constructing the action concepts in a dataset-agnostic way.
- I could not find the ablation of the uniform distribution prior regularization. It is important to understand the influence of that prior on the training. Could the authors please include this?
- The writing could be significantly improved.

**Questions:**

in line 79, the authors probably meant “self-supervised learning” instead of “semi-supervised”? Semi-supervised assumes you have access to some labeled data.

**Limitations:**

It is not very clear whether knowing the text action concepts is important for the method to work well. This may be a potential limitation.

---

> ### Author Rebuttal · Authors · 2023-08-09
>
> We thank the reviewer for the positive feedback and address all suggestions below.
>
> 1. `Use dataset-agnostic action concepts:` We take the reviewers advice and develop two alternate strategies to construct action concepts in a dataset-agnostic way: GPT based category generation (LSS-B) and image VQA based labels (LSS-C). These methods have no reliance on dataset textual information and obtain results (rebuttal PDF Table 1, 2) on par with our default setting (LSS-A). We also include these results for linear probing (top) and zero-shot (bottom) top-1 accuracy below for quick reference.
>
>     |    Method    | ITP | HMDB-51 |  UCF-101 |
>     |:------------:|:---:|:----:|:----:|
>     | LSS-A (ours) | yes | 69.2 | 91.0 |
>     | LSS-B (ours) | yes | **69.4** | **91.1** |
>     | LSS-C (ours) | yes | 69.1 | 90.8 |
>
>     | Method | Action Labels    | HMDB | UCF  |
>     |--------|------------------|------|------|
>     | CLIP   | -                | 47.2 | 70.3 |
>     | Ours   | K400+U+H (LSS-A)     | 49.5 | 72.0 |
>     | Ours   | GPT labels (LSS-B)   | 50.2 | 73.8 |
>     | Ours   | I-VLM labels (LSS-C) | **51.4** | **74.2** |
>
>     This highlights how LSS can operate without knowing the textual action concepts of the training or downstream datasets.
>
> 2. `Add uniform distribution prior (UDP) ablation:` This UDP regularization is crucial for stable training. Without it the pre-training stage leads to collapse. We include ablations here and in rebuttal PDF (Table 3).
>
>     | Method      | HMDB | UCF  |
>     |-------------|------|------|
>     | Default LSS | 48.4 | 71.1 |
>     | w/o UDP      | 33.4 | 54.3 |
>
>
> 3. `Fix typos and writing:`
> Thank you for pointing out the typo on L79 - we will fix that to be *self-supervised learning*. We will also revise our final manuscript further to improve our writing.

---

> > ### Comment · Reviewer_aWeV · 2023-08-16
> > **Response**
> >
> > I thank the authors for their response.
> >
> > I think point 1 makes the paper look stronger and point 2 gives important intuition about the problem. I advise to include them in the final paper.
> >
> > Otherwise, the rebuttal answers all my questions. I keep my score as week accept.

---

> > > ### Author Response · Authors · 2023-08-16
> > >
> > > We thank the reviewer for their comments and all useful feedback. We will update our final manuscript to reflect all these proposed modifications.

---

### Official Review · Reviewer_mCow · 2023-07-04

**Soundness:** 3 good
**Presentation:** 3 good
**Contribution:** 2 fair
**Rating:** 5
**Confidence:** 4

**Summary:**

The paper introduces a language-tied self-supervised learning approach to adapt an image CLIP model to the video domain. The method employs two video-specific self-supervised learning objectives: concept distillation and concept alignment, for training the model. The authors showcase that the proposed method achieves good zero-shot and linear probing performance on three action recognition benchmarks.

**Strengths:**

1. The motivation behind using language for video self-supervised learning is good as it addresses the existing challenges associated with video datasets and holds the potential to offer effective solutions.
2. The paper presents good zero-shot performance
3. The paper is well written and the overall message is well understood.

**Weaknesses:**

1. While this paper refers to itself as a self-supervised method, it relies significantly on labeled information. For instance, the text classifier utilizes language embeddings derived from the dataset categories. Additionally, the construction of action concept spaces and category concept spaces takes into account the awareness of dataset categories. This information leak introduces a potential unfairness in comparing the proposed self-supervised learning method with other SSL methods.
2. Consider adding experiments that involve removing the awareness of dataset-level labels to align with other SSL methods.
3. It is not fair to directly compare the proposed method, which utilizes pre-trained CLIP weights, with other SSL methods that are trained from scratch. This distinction should be discussed, and the attribute of pre-training should be added to Table 1.

**Questions:**

One of the main issues with this paper is the misleading setting employed by LSS. As mentioned in the weaknesses section, LSS is not a purely self-supervised method as it requires some dataset-level labels and also uses CLIP pre-trained weights. It is important to differentiate this setting from traditional SSL settings and explain the practical utility of the proposed LSS setting in applications.

**Limitations:**

The authors have addressed the limitations.

---

> ### Author Rebuttal · Authors · 2023-08-09
>
> We thank the reviewer for the helpful comments and address all concerns below.
>
> 1. `Experiments removing dataset-level label awareness:` We eliminate the need for dataset-level labels in 2 additional variants (GPT based category generation and image VQA based labels) and run experiments for these variants. Results (presented in rebuttal PDF Table 1, 2) demonstrate how LSS can be trained without any textual information from video datasets, removing all dataset-level label awareness. We also include these results for linear probing (top) and zero-shot (bottom) top-1 accuracy below.
>     |    Method    | ITP | HMDB-51 |  UCF-101 |
>     |:------------:|:---:|:----:|:----:|
>     | LSS-A (ours) | yes | 69.2 | 91.0 |
>     | LSS-B (ours) | yes | **69.4** | **91.1** |
>     | LSS-C (ours) | yes | 69.1 | 90.8 |
>
>     | Method | Action Labels    | HMDB | UCF  |
>     |--------|------------------|------|------|
>     | CLIP   | -                | 47.2 | 70.3 |
>     | Ours   | K400+U+H (LSS-A)     | 49.5 | 72.0 |
>     | Ours   | GPT labels (LSS-B)   | 50.2 | 73.8 |
>     | Ours   | I-VLM labels (LSS-C) | **51.4** | **74.2** |
>
>
> 2. `Fair comparison:` We update Table 1 (in main rebuttal PDF) to distinguish methods using CLIP pre-training and include more methods using such pre-training.  LSS performs the best among prior work using CLIP pre-training, showcasing our unique strengths.
>
> 3. `Differentiate from traditional SSL:` LSS uses image-text pre-training additionally compared to traditional SSL approaches. However, LSS contains zero-shot capabilities unlike prior SSL works. This unique ability creates multiple practical utilities of LSS over prior SSL works.  We will update our final manuscript to discuss this clearer.

---

> > ### Comment · Reviewer_mCow · 2023-08-19
> >
> > Thanks for your response. However, the rebuttal does not address my major concerns. Even though two additional variations have been introduced (LSS-B and LSS-C), it appears that they still rely on dataset-level label awareness. To illustrate, the i-LVM labels continue to be derived from the visual contents within the datasets. I kindly request clarification if my interpretation is inaccurate. Furthermore, I recommend that the authors provide comprehensive explanations regarding the methodology employed for generating these newly introduced labels.

---

> > > ### Author Response · Authors · 2023-08-19
> > > **No reliance on dataset-level labels**
> > >
> > > We apologize for the lack of clarity on our part. LSS-B uses *no dataset level information at all* - it contains generic action labels. LSS-C uses captioning models (trained on images similar to CLIP, no access to videos) to generate labels, and *accesses only videos in training dataset* (same videos used for SSL training). We explain in detail below.
> > >
> > > For LSS-B, we use GPT to generate a large set of action labels. We first prompt GPT to categorize all common human actions / activities into 20 groups. For each group, we again ask GPT to generate at least 100 visually diverse action categories. These are all collected to create a set of 2000 action labels. We then use projections of these labels in CLIP text-encoder representation space to eliminate labels of high semantic similarity, achieving only 1000 diverse action categories. So our **1000 action categories for LSS-B are generic, and not tied to any of our training datasets**. This experiment demonstrate the scalability of our approach without accessing annotated dataset labels.
> > >
> > > For LSS-C, we generate a label set using only videos from the training dataset. We use PCA based clustering to identify 2000 representative videos from a randomly sampled subset (50,000) of our training dataset and then use image-captioning models on video center frames to generate a diverse set of 2000 action labels. This is further reduced to 500 eliminating labels that are similar in feature space of the CLIP text encoder. In this case, our generated **labels are tied to the training dataset, but uses no annotated category labels**. We use only the videos (same videos used for SSL training) and an image-to-text captioning model (trained on images) to generate our label set.
> > >
> > > These generated label sets are then used (in place of ground truth category labels from dataset) to construct our proposed action concept spaces . They are treated as the action concept set for the rest of our SSL training.
> > >
> > > We will ensure to highlight these details better in our revised manuscript.

---

> > > > ### Comment · Reviewer_mCow · 2023-08-20
> > > >
> > > > Thank you for clarifying. My major concern has been resolved. I have raised my score to borderline accept.

---

> > > > > ### Author Response · Authors · 2023-08-20
> > > > >
> > > > > We thank the reviewer again for all feedback and positive comments. We will update our final version accordingly.

---

### Official Review · Reviewer_WHFS · 2023-07-06

**Soundness:** 2 fair
**Presentation:** 3 good
**Contribution:** 2 fair
**Rating:** 5
**Confidence:** 4

**Summary:**

The paper presents a method to adapt a vision-language model (CLIP) to represent videos.
The method extends the image encoder of CLIP to a video encoder via factored space-time attention.
The paper introduces a self-distillation-based objective to adapt the video encoder to train on unlabelled videos (no video captions are required).
This self-distillation is performed in a so-called "action concept space" which results from projecting visual embeddings into the space spanned by 0-shot action classifiers or text embeddings of action descriptions.
The approach is evaluated on Kinetics 400, UCF101, and HMBD in 0-shot and linear action classification experiments.

**Strengths:**

- The adaptation approach does not require additional video-level captions (only knowledge of action categories in the training data is required)
- The method performs well in "zero-shot" and linear probing experiments in the considered action datasets
- The use of fixed projections obtained through 0-shot classifiers for distillation is interesting and seems effective when the classes observed in transfer are known
- The paper is overall well written and the method well presented

**Weaknesses:**

- The experiments are limited to 0-shot and linear classification of the same action classes also observed and used in the adaption training. It is unclear if the method can generalize to new actions (ones not used to define the action concept spaces)
- The experiments also lack any text-to-video retrieval benchmarks. These would be essential to demonstrate the claim that the method "preserves and improves the strengths of CLIP ... for video operation" L49
- The comparisons to most methods in Table 1 are unfair since they are all fully self-supervised (no captions used in pre-training), and most are trained exclusively on videos (much less effective training data). Also, it is unclear why many methods are included without any performance numbers. Instead, it would be better to compare to other CLIP-based methods as in Tab 2 in this benchmark as well
- The difference in Tab 2 and 3 suggests the method is very sensitive to the set of actions used to define the projection space

**Questions:**

I would appreciate it if the authors could address my concerns listed in the weaknesses above. Most importantly, I would like to know:
- How does the model perform for text-to-video retrieval?
- How does the model generalize to actions not used during pre-training? For example, what if Tab 5 only consists of K400 classes?

Additionally, I'm wondering:
- What is the importance of w_s (Eq 5)?

**Limitations:**

Some of the limitations I see (see weaknesses) have not been addressed in the paper.

---

> ### Author Rebuttal · Authors · 2023-08-09
>
> We thank the reviewer for the detailed feedback and address all comments below.
>
> 1. `Text-to-video retrieval:` We run experiments on MSR-VTT text-to-video retrieval benchmark to demonstrate how LSS improves over our baseline CLIP. The performance increase is significant and consistent to our prior results.
>
>     | Method | R@1  | R@5  | R@10 |
>     |--------|------|------|------|
>     | CLIP   | 30.6 | 54.4 | 64.3 |
>     | LSS    | 33.8 | 58.2 | 70.3 |
>
> 2. `Generalize to actions not used during pre-training:` We run 3 new experiments to demonstrate how LSS generalizes to unseen actions. First, pre-training on K400 labels only (action classes overlapping with UCF/HMDB are removed here) with UCF/HMDB evaluation is reported in Table 2 (rebuttal PDF) and below (zero-shot top-1 accuracy).
>
>
>     | Method | Action Labels | HMDB | UCF  |
>     |--------|---------------|------|------|
>     | CLIP   | -             | 47.2 | 70.3 |
>     | Ours   | K400 only (w/o U, H)     | 48.4 | 71.1 |
>     | Ours   | K400+U+H (original)     | 49.5 | 72.0 |
>
>     Next we show results for experiments we run on 2 variants introduced in main rebuttal, LSS-B and LSS-C, that use **no dataset textual labels** for pre-training. These results are reported in Table 1 & 2 (rebuttal PDF) and surpass our default setting. We also report these below (zero-shot top-1 accuracy).
>
>     | Method | Action Labels    | HMDB | UCF  |
>     |--------|------------------|------|------|
>     | CLIP   | -                | 47.2 | 70.3 |
>     | Ours   | K400+U+H (original)     | 49.5 | 72.0 |
>     | Ours   | GPT labels (LSS-B)   | 50.2 | 73.8 |
>     | Ours   | I-VLM labels (LSS-C) | 51.4 | 74.2 |
>
>
> 3. `Importance of w_s:` This term represents confidence of the target concept space projection for a given sample. Since each basis of the concept space corresponds to action categories, if a sample is more aligned to a single basis, we assume higher confidence in that sample, leading to higher w_s value (max element of softmax normalized projected vector). And the reverse when a sample is equally aligned to a number of basis axes.
> Since each sample during training is a clip sampled from a video (which covers a temporal crop of video), our intuition for this weight is to act as a way of prioritizing more important clips over the less important ones. We also include ablation for w_s below.
>
>     | Method      | HMDB | UCF  |
>     |-------------|------|------|
>     | Default LSS | 48.4 | 71.1 |
>     | w/o w_s     | 47.2 | 70.3 |
>
>
> 4. `Unfair comparison in Table 1:` We update Table 1 (please see main rebuttal PDF) to include comparisons to CLIP based methods using image-text pre-training (ITP). Results show how our proposed LSS performs significantly better than prior ITP approaches and retains strengths of CLIP. Missing performance for some methods was a latex typo - we have fixed that, thanks for the pointer! We add updated Table 1 below too (linear probing top-1 accuracy).
>
>     |    Method    | ITP | HMDB-51 |  UCF-101 |
>     |------------|:---:|:----:|:----:|
>     |      SVT     |  no | 57.8 | 90.8 |
>     |   VideoMAE   |  no | 60.3 | 84.7 |
>     |    MERLOT    | yes | 55.4 | 80.1 |
>     |     VATT     | yes | 66.4 | 87.6 |
>     |     TVTS     | yes | 58.4 | 83.4 |
>     |    LaViLa    | yes | 61.5 | 88.1 |
>     | LSS-A (ours-original) | yes | 69.2 | 91.0 |
>     | LSS-B (ours-new) | yes | **69.4** | **91.1** |
>     | LSS-C (ours-new) | yes | 69.1 | 90.8 |

---

> > ### Comment · Reviewer_WHFS · 2023-08-20
> >
> > I have read the rebuttal and the other reviews. I appreciate the comprehensive author's response and novel results. The new results make the paper much more convincing and largely resolve my concerns. I'm happy to increase my rating.

---

> > > ### Author Response · Authors · 2023-08-20
> > >
> > > We thank the reviewer for their consideration and highly useful feedback. We will update our final manuscript accordingly with all additional material from rebuttal.

---

### Official Review · Reviewer_Cb1q · 2023-07-06

**Soundness:** 3 good
**Presentation:** 3 good
**Contribution:** 3 good
**Rating:** 6
**Confidence:** 4

**Summary:**

The paper proposes a novel language-based self-supervised learning framework (LSS) for video representation learning. It extends the self-distillation based SSL approaches like BYOL and SimSiam by replacing the randomly initialized project network by the text classifier defined by language embeddings extracted from the image CLIP text encoder. With two novel self-supervised learning objectives, the pretrained video model retains and improves transferability and generality of image CLIP representations better in comparison to existing video SSL methods. LSS achieves state-of-the-art results under linear probing settings and competitive zero-shot transfer performances on HMDB-51 and UCF-101.

**Strengths:**

+ The paper is overall well written and easy to follow.
+ The idea of replacing the project network with the text classifier defined by image CLIP embeddings is really interesting and makes sense. The video representation learning suffers from relatively expensive and noisy annotations; we can easily distill knowledges obtained from abundant and diverse image-based datasets using LSS.
+ The experimental results show the high transferability and generality of the proposed method, LSS.

**Weaknesses:**

- LSS uses action categories of Kinetics-400, UCF-101 and HMDB-51 for defining the action category and description concept spaces. However, if those action concept spaces can be used only for pretraining on one of the three datasets used for concept space construction, this means that we can only use manually annotated action recognition datasets for pretraining. This limits the data scalability of LSS. The authors should provide experimental results with pretraining on more general, non-labeled video data using the same action concept spaces. Otherwise, for using abundant web videos without labeling, the authors should come up with the action space construction method without manually annotated action categories.
- Please fix the typos; e.g., the l2 norm, not the squared l2 norm, should be used in Eq. (1) and (3) for normalization.

**Questions:**

The key idea of the paper, replacing the project network with the text classifier really makes sense and is very interesting. I will lean towards acceptance if the authors address my concerns in the rebuttal.

**Limitations:**

The authors have adequately addressed the limitations and potential societal impact of the work.

---

> ### Author Rebuttal · Authors · 2023-08-09
>
> We thank the reviewer for positive comments and helpful feedback. We address all concerns below.
>
> 1. `Pre-training without dataset labels:` As we also described in the main rebuttal, we run 2 additional experiments for variants of our method that use NO textual labels from any datasets. Results (LSS-B & LSS-C in Tables 1, 2 of rebuttal PDF) showcase equally strong performance by these variants. We also report the same linear probing top-1 accuracy below.
>     |    Method    | ITP | HMDB-51 |  UCF-101 |
>     |------------------|:---:|:----:|:----:|
>     | LSS-A (ours-original) | yes | 69.2 | 91.0 |
>     | LSS-B (ours-new) | yes | **69.4** | **91.1** |
>     | LSS-C (ours-new) | yes | 69.1 | 90.8 |
>
>     Zero-shot top-1 accuracy compared to a CLIP baseline are also reported below again for quick reference.
>     | Method | Action Labels    | HMDB | UCF  |
>     |--------|------------------|------|------|
>     | CLIP   | -                | 47.2 | 70.3 |
>     | Ours   | K400+U+H (LSS-A)     | 49.5 | 72.0 |
>     | Ours   | GPT labels (LSS-B)   | 50.2 | 73.8 |
>     | Ours   | I-VLM labels (LSS-C) | **51.4** | **74.2** |
>
>     These results indicate that LSS can be used with pre-training video datasets that contain no action annotations.
>
> 2. `Label-free action space construction:` The LSS-B variant uses GPT-3 to generate a set of 2000 common activity labels. This is reduced to 1000 by eliminating labels that are similar in feature space of the CLIP text encoder and then used to construct the action space. The LSS-C variant generates a label set using only videos from the training dataset. We use PCA based clustering to identify 2000 representative videos from a randomly sampled subset of our training dataset and then use image-captioning models on video center frames to generate a diverse set of 2000 action labels. This is further reduced to 500 eliminating labels that are similar in feature space of the CLIP text encoder and then used to construct the action space. We will elaborate these details further in our revision.
>
>
> 3. `Typos:` Thank you for pointing out these typos - we have fixed them.

---

> > ### Comment · Reviewer_Cb1q · 2023-08-20
> >
> > I appreciate the efforts of the authors to provide rebuttals. The authors addressed most of the reviewers' concerns, and especially Tables 1 and 2 make the proposed approach more stronger. Please add them in the final draft. I will raise my score to week accept.

---

> > > ### Author Response · Authors · 2023-08-20
> > >
> > > We thank the reviewer for the useful feedback and positive comments. We will update the final draft with all these details.

---

### Author Rebuttal · Authors · 2023-08-08

We thank all reviewers for positive comments: results show high transferability and generality of method (R-Cb1q); interesting and seems effective for transfer on known classes (R-WHFS); presents good zero-shot performance, holds the potential to offer effective solutions (R-mCow); tackles an important problem, extensive evaluation and strong results (R-aWeV); impressive performance on multiple datasets, ablations demonstrate effectiveness (R-vhEc).


We discuss our two main modifications in response to concerns raised by reviewers below.

1. `Concern: Reliance on textual action categories of datasets`
  * We run experiments on two additional variants of LSS that do NOT rely on any dataset labels. They utilize different concept spaces (variants B & C in rebuttal PDF Table 1) that use no textual class information from training or downstream datasets. These obtain similar performance (superior to prior work) in both linear probing and zero-shot settings, highlighting how **proposed LSS can operate without access to textual action categories of datasets**.


2. `Concern: Fairer comparison against prior SSL work`
  * Following suggestions from the reviewers, we modify Table 1 (in rebuttal PDF) to explicitly highlight our use of image-text pretraining (ITP). We also include comparisons to related works that use image-text pretraining. Results show how our proposed LSS performs significantly better than these image-text pretraining approaches.
  * We also reiterate the additional zero-shot capabilities of our method that traditional SSL do not possess.

Please refer to attached PDF for tables. Further rebuttals are written as responses to each review addressing the concerns raised by reviewers.

---

### Decision · Program_Chairs · 2023-09-21

**Decision:**

Accept (poster)

**Comment:**

All of the reviewers recommended the paper. They appreciated the strong zero-shot results and well-written justifications in the paper. The authors should be sure to update the paper based on the rebuttals.